



# FABM-NflexPD 2.0: Testing an Instantaneous Acclimation Approach for Modelling the Implications of Phytoplankton Eco-physiology for the Carbon and Nutrient cycles

Onur Kerimoglu[1], Markus Pahlow[2], Prima Anugerahanti[3,a], and Sherwood Lan Smith[3]

[1]Institute for Chemistry and Biology of the Marine Environment, University of Oldenburg, Oldenburg, Germany
[2]GEOMAR Helmholtz Centre for Ocean Research Kiel, Kiel, Germany
[3]Earth SURFACE Research Center, Research Institute for Global Change, JAMSTEC, Yokosuka, Japan
[a]Present address: Dept. of Earth, Ocean and Ecological Sciences, School of Environmental Sciences, University of Liverpool, Liverpool, United Kingdom

**Correspondence:** Onur Kerimoglu (kerimoglu.o@gmail.com)

**Abstract.** The acclimative response of phytoplankton, which adjusts their nutrient and pigment content in response to changes in ambient light, nutrient levels, and temperature, is an important determinant of observed chlorophyll distributions and biogeochemistry. Acclimative models typically capture this response and its impact on the C:nutrient:Chl ratios of phytoplankton by explicitly resolving the dynamics of these constituents of phytoplankton biomass. The Instantaneous Acclimation (IA) approach only requires resolving the dynamics of a single tracer and calculates the elemental composition assuming instantaneous local equilibrium. IA can capture the acclimative response without substantial loss of accuracy in both 0D box models and spatially explicit 1D models. A major draw-back of IA so far has been its inability to maintain mass balance for the elements with unresolved dynamics. Here we extend the IA model to capture both C and N cycles in a 0D setup, which requires analytical derivation of additional flux terms to account for the temporal changes in cellular N quota, $Q$. We present extensive tests of this model, with regard to the conservation of total C an N, and its behavior in comparison to an otherwise equivalent, fully explicit Dynamic Acclimation (DA) variant, under idealized conditions with variable light and temperature. We also demonstrate a modular implementation of this model in the Framework for Aquatic Biogeochemical Modelling (FABM), which facilitates modelling competition between an arbitrary number of different acclimative phytoplankton types. In a 0D setup, we did not find evidence for computational advantages of the IA approch over the DA variant. In a spatially explicit setup, performance gains may be possible, but would require modifying the physical-flux calculations to account for spatial differences in $Q$ between model grid cells.



## 1   Introduction

Elemental stoichiometry and pigment density of phytoplankton vary widely across environmental conditions, at both the physiological (e.g., Garcia et al., 2016), and the community level (e.g., Moreno and Martiny, 2018). The physiological flexibility is
driven by an acclimative re-adjustment of cellular machinery to changes in the availability in nutrients and light, and the fact that the various cellular functions have competing requirements for resources, an example being enzymes, rich in N (Geider and La Roche, 2002), needed for nutrient uptake and photosynthesis. The systematic differences in cellular composition between species may be explained by the typical composition of some species being better or worse suited than that of others for a given resource regime (Klausmeier et al., 2004; Arrigo, 2005; Burson et al., 2016).

The potential relevance of such variability in the cellular composition of phytoplankton for biogeochemical cycles has been recognized decades ago (Redfield, 1934, 1958), and evidence has been building ever since (Lenton and Klausmeier, 2007; Bonachela et al., 2016; Pahlow et al., 2020). Accounting for the acclimative capacity of phytoplankton in models is relevant for predicting the response of ecosystems to environmental change (Kwiatkowski et al., 2018; Kerimoglu et al., 2018) and for model performance (Ayata et al., 2013; Kerimoglu et al., 2017; Chen and Smith, 2018). It also can endow models with desirable
properties, such as improved model portability (Anugerahanti et al., 2021). However, mechanistic acclimative models typically require additional state variables, usually one for chlorophyll and one for each of the resolved nutrients (e.g., Geider et al., 1998; Flynn, 2003), but possibly even more (Bonachela et al., 2013; Wirtz and Kerimoglu, 2016; Inomura et al., 2020).

However, additional state variables can increase the computational costs significantly in spatially explicit setups (Fulton et al., 2003), especially for models with 100s of phytoplankton groups (e.g., Follows et al., 2007; Dutkiewicz et al., 2020). As
a potential remedy to this problem, the 'Instantaneous Acclimation' (IA) approach can be used, where the changes in cellular composition are not dynamically tracked, but adjust instantaneously to the resource environment. For instance, the FlexPFT model (Smith et al., 2016) follows an IA approach, where the Chl:C and C:N ratios instantaneously assume the optimal ratios for balanced growth (Pahlow et al., 2013). Ward (2017) compared a fully explicit, classical Caperon/Droop model (Caperon, 1968; Droop, 1968) to its IA counterpart, and found that across a range of environmental settings, the predictions of the two
approaches matched closely in a 0D setup. Kerimoglu et al. (2021) introduced FABM-NflexPD 1.0, a 1D setup of the FlexPFT model in the Framework of Aquatic Biogeochemical Models (FABM, Bruggeman and Bolding, 2014) and showed that the predictions of the IA variant of the model largely matched those of the fully explicit 'Dynamic Acclimation' (DA) counterpart, except for minor differences during the transitions from winter to spring and from autumn to winter.

FABM-NflexPD 1.0 only tracks N, which may be sufficient for some ecological applications, but not for applications that
require mass balance for (multiple) nutrients and carbon. Here we introduce FABM-NflexPD 2.0, which tracks both C and N. We present a detailed description of a C-based version, which is extended to account also for N fluxes resulting from instantaneous changes in cell quotas, such that mass balance is maintained for both C and N. We evaluate the consistency and robustness of the model by means of the following formal tests:

**T1** assessment of the conservation of C and N in a simplified version of the model in a 0D setup, where temperature and day
length are held constant, and light is provided as a sinusoidal function of time of year



**T2** test of the IA approach in a more realistic setup, where temperature and day length also vary over time, while also accounting for light attenuation by phytoplankton, and comparison against a fully explicit DA variant

**T3** simulation with multiple phytoplankton groups

**T4** simulation in an open system where N and C are not conserved

## 2   Model Description

FABM-NflexPD 2.0 differs from FABM-NflexPD 1.0 (K21) mainly by tracing the dissolved inorganic carbon pool, DIC, such that now the model is ideally able to conserve both the total C and N in the system, and not only N as in K21. Moreover, we consider dilution/mixing and sinking terms for simulating an open system, such as a chemostat or a surface mixed layer (SML). As in K21, we consider here a DA variant to compare against the IA variant, but not a fixed stoichiometry variant unlike in K21. As a final difference, for the IA, we trace the C content in phytoplankton with a state variable, instead of the N content as in K21. The rates of change of the state variables are:

$$\frac{\mathrm{d}\,\mathrm{Phy_C}}{\mathrm{d}t} = F_{\mathrm{DIC-Phy_C}} - F_{\mathrm{Phy_C-Det_C}} - D \cdot \mathrm{Phy_C} \tag{1a}$$

$$\frac{\mathrm{d}\,\mathrm{Phy_N}}{\mathrm{d}t} = \underbrace{F_{\mathrm{DIN-Phy_N}}}_{\text{Net uptake}} - \underbrace{F_{\mathrm{Phy_N-Det_N}}}_{\text{Mortality}} - \underbrace{D \cdot \mathrm{Phy_N}}_{\text{Dilution}} \tag{1b}$$ $\{\mathrm{DA}\}$

$$\frac{\mathrm{d}\,\mathrm{Det_C}}{\mathrm{d}t} = F_{\mathrm{Phy_C-Det_C}} - F_{\mathrm{Det_C-DOC}} - D \cdot \mathrm{Det_C} - \frac{w_{\mathrm{Det}}}{H_{\mathrm{SML}}} \cdot \mathrm{Det_C} \tag{2a}$$

$$\frac{\mathrm{d}\,\mathrm{Det_N}}{\mathrm{d}t} = F_{\mathrm{Phy_N-Det_N}} - \underbrace{F_{\mathrm{Det_N-DON}}}_{\text{Hydrolysis}} - D \cdot \mathrm{Det_N} - \underbrace{\frac{w_{\mathrm{Det}}}{H_{\mathrm{SML}}} \cdot \mathrm{Det_N}}_{\text{Sinking}} \tag{2b}$$

$$\frac{\mathrm{d}\,\mathrm{DOC}}{\mathrm{d}t} = F_{Det_C-DOC} - F_{DOC-DIC} - D \cdot \mathrm{DOC} \tag{3a}$$

$$\frac{\mathrm{d}\,\mathrm{DON}}{\mathrm{d}t} = F_{Det_N-DON} - \underbrace{F_{DON-DIN}}_{\text{Remineralization}} - D \cdot \mathrm{DON} \tag{3b}$$

$$\frac{\mathrm{d}\,\mathrm{DIC}}{\mathrm{d}t} = F_{\mathrm{DOC-DIC}} - F_{\mathrm{DIC-Phy_C}} - D \cdot (\mathrm{DIC} - \mathrm{DIC_{in}}) \tag{4a}$$

$$\frac{\mathrm{d}\,\mathrm{DIN}}{\mathrm{d}t} = F_{\mathrm{DON-DIN}} - F_{\mathrm{DIN-Phy_N}} - D \cdot (\mathrm{DIN} - \mathrm{DIN_{in}}) \tag{4b}$$

where $F_{x-y}$ is the flux from $x$ to $y$, where $x$ and $y$ are state variables, except for $\mathrm{Phy_N}$ in the IA variant, which is defined as:

$$\mathrm{Phy_N} = \mathrm{Phy_C} \cdot Q \tag{5}$$ $\{\mathrm{IA}\}$

where $Q$ is the phytoplankton N quota (N:C ratio). Dilution and sinking terms describe fluxes in and out of the system, and are non-zero only for the test T4 (see below).



## 2.1  C and N content of Phytoplankton

As mentioned above, in this, C-based, version of the IA model variant, only the C content of phytoplankton ($\text{Phy}_\text{C}$) is dynamically tracked via Eq. (1a), whereas $\text{Phy}_\text{N}$ is defined as a function of $Q$ in Eq. (5). $Q$ adjusts instantaneously to its optimal value for balanced growth, as determined by nutrient uptake in the protoplast, $\hat{V}_\text{N}$, and net photosynthesis in the chloroplast, $\hat{\mu}_\text{net}$ (Eq. 10 in K21):

$$Q = \frac{Q_0}{2}\left[1 + \sqrt{1 + \frac{2}{Q_0(\hat{\mu}_\text{net}/\hat{V}_\text{N} + \zeta_\text{N})}}\right] \qquad \text{\{IA\}} \quad (6)$$

where $Q_0$ and $\zeta_\text{N}$ are the subsistence N quota, and cost of N uptake, respectively (Table 1). The first term in Eq. (1a), $F_{\text{DIC}-\text{Phy}_\text{C}}$, represents net phytoplankton growth:

$$F_{\text{DIC}-\text{Phy}_\text{C}} = \mu \cdot \text{Phy}_\text{C} \qquad (7)$$

The second term in Eq. (1a), $F_{\text{Phy}_\text{C}-\text{Det}_\text{C}}$, represents the mortality of phytoplankton:

$$F_{\text{Phy}_\text{C}-\text{Det}_\text{C}} = m_\text{C} \cdot \text{Phy}_\text{C}^2 \qquad (8)$$

Its N counterpart is found by multiplication with $Q$:

$$F_{\text{Phy}_\text{N}-\text{Det}_\text{N}} = F_{\text{Phy}_\text{C}-\text{Det}_\text{C}} \cdot Q \qquad (9)$$

where $m_\text{C}$ [$\text{m}^3 \, \text{mmolC}^{-1} \, \text{d}^{-1}$] is the C-based specific mortality rate. The hydrolysis and remineralization fluxes are calculated as first-order reactions:

$$F_{\text{Det}_X-\text{DO}X} = r_\text{hyd} \cdot \text{Det}_X \, , \quad X \in \{\text{C, N}\} \qquad (10)$$

$$F_{\text{DO}X-\text{DI}X} = r_\text{rem} \cdot \text{DO}X \, , \quad X \in \{\text{C, N}\} \qquad (11)$$

## 2.2  N fluxes between DIN and Phytoplankton

For tracking the $\text{Phy}_\text{N}$ for the DA variant (Eq. 1b), and the DIN for both variants (Eq. 4b), the flux from DIN to $\text{Phy}_\text{N}$ needs to be known. As in K21, for the DA variant, it is simply the product of a specific uptake rate and the phytoplankton C biomass:

$$F_{\text{DIN}-\text{Phy}_\text{N}} = V_\text{N} \cdot \text{Phy}_\text{C} \qquad \text{\{DA\}} \quad (12)$$

For the IA variant, the exact value of this flux is unknown due to the non-existent $\text{Phy}_\text{N}$ pool, and the corresponding flux. By substituting $\text{Phy}_\text{N}$ with $\text{Phy}_\text{C} \cdot Q$, and applying the product rule we get:

$$\frac{\text{d}\,\text{Phy}_\text{N}}{\text{d}\,t} = \frac{\text{d}\,(\text{Phy}_\text{C} \cdot Q)}{\text{d}\,t} = \frac{\text{d}\,\text{Phy}_\text{C}}{\text{d}\,t} \cdot Q + \text{Phy}_\text{C} \cdot \frac{\text{d}\,Q}{\text{d}\,t} \qquad (1b.2)$$

here, the first term reflects the N equivalent of the change in $\text{Phy}_\text{C}$, i.e., Eq. (1a), and the second term describes the effect of the change in quota over time due to imbalances between C and N uptake. Substituting Eq. (1a) in Eq. (1b.2):

$$\frac{\text{d}\,\text{Phy}_\text{N}}{\text{d}\,t} = \left[F_{\text{DIC}-\text{Phy}_\text{C}} \cdot Q - F_{\text{Phy}_\text{C}-\text{Det}_\text{C}} \cdot Q - D \cdot \text{Phy}_\text{C} \cdot Q\right] + \text{Phy}_\text{C} \cdot \frac{\text{d}\,Q}{\text{d}\,t} \qquad (1b.3)$$





and plugging Eqs. (7) and (9) into Eq. (1b.3) yields:

$$\frac{\mathrm{d}\,\mathrm{Phy_N}}{\mathrm{d}\,t} = \left[\mu \cdot \mathrm{Phy_C} \cdot Q - F_{\mathrm{Phy_N - Det_N}} - D \cdot \mathrm{Phy_C} \cdot Q\right] + \mathrm{Phy_C} \cdot \frac{\mathrm{d}\,Q}{\mathrm{d}\,t} \tag{1b.4}$$

In Eq. (1b.4), $\mu \cdot \mathrm{Phy_C} \cdot Q$ can be identified with $F_{\mathrm{DIN - Phy_N}}$ in the IA variant because $V_\mathrm{N} = \mu \cdot Q$ for balanced growth. Following Smith et al. (2016), we assign that the last term to $F_{\mathrm{DIN - Phy_N}}$ too, yielding a re-definition of $F_{\mathrm{DIN - Phy_N}}$ for the IA variant:

$$F_{\mathrm{DIN - Phy_N}} = \mathrm{Phy_C}\left[\mu Q + \frac{\mathrm{d}\,Q}{\mathrm{d}\,t}\right] \tag{\{IA\} (12.2)}$$

which essentially redirects part of the fluxes associated with $\mathrm{Phy_N}$ to DIN. Plugging this into Eq. (4b), and recognizing that

$\frac{\mathrm{d}\,Q}{\mathrm{d}\,t}$ consists of partial derivatives with respect to DIN, daily average irradiance, $\bar{I}$, fractional daylength, $L_\mathrm{D}$, and temperature, $T$ (see Appendix A):

$$\frac{\mathrm{d}\,\mathrm{DIN}}{\mathrm{d}\,t} = F_{\mathrm{DON - DIN}} - \mathrm{Phy_C}\left[\mu Q + \frac{\mathrm{d}\,Q}{\mathrm{d}\,t}\right] - D \cdot (\mathrm{DIN} - \mathrm{DIN_{in}})$$

$$= F_{\mathrm{DON - DIN}} - \mathrm{Phy_C}\left[\mu Q + \frac{\partial Q}{\partial \mathrm{DIN}}\frac{\mathrm{d}\,\mathrm{DIN}}{\mathrm{d}\,t} + \frac{\partial Q}{\partial \bar{I}}\frac{\mathrm{d}\,\bar{I}}{\mathrm{d}\,t} + \frac{\partial Q}{\partial L_\mathrm{D}}\frac{\mathrm{d}\,L_\mathrm{D}}{\mathrm{d}\,t} + \frac{\partial Q}{\partial T}\frac{\mathrm{d}\,T}{\mathrm{d}\,t}\right] - D \cdot (\mathrm{DIN} - \mathrm{DIN_{in}}) \tag{\{IA\} (4b.2)}$$

and reorganizing the $\frac{\mathrm{d}\,\mathrm{DIN}}{\mathrm{d}\,t}$ on the right hand side:

$$\frac{\mathrm{d}\,\mathrm{DIN}}{\mathrm{d}\,t} = \frac{F_{\mathrm{DON - DIN}} - \mathrm{Phy_C}\left[\mu Q + \dfrac{\partial Q}{\partial \bar{I}}\dfrac{\mathrm{d}\,\bar{I}}{\mathrm{d}\,t} + \dfrac{\partial Q}{\partial L_\mathrm{D}}\dfrac{\mathrm{d}\,L_\mathrm{D}}{\mathrm{d}\,t} + \dfrac{\partial Q}{\partial T}\dfrac{\mathrm{d}\,T}{\mathrm{d}\,t}\right] - D \cdot (\mathrm{DIN} - \mathrm{DIN_{in}})}{1 + \mathrm{Phy_C}\dfrac{\partial Q}{\partial \mathrm{DIN}}} \tag{\{IA\} (4b.3)}$$

The partial derivatives of $Q$ with respect to DIN, $\bar{I}$, $L_\mathrm{D}$ and $T$ are obtained by canonical application of the chain rule, as detailed in Appendix A. The final terms required in Eq. (4b.3) are the changes in $\bar{I}$, $L_\mathrm{D}$ and $T$ over time, i.e., $\mathrm{d}\,\bar{I}/\mathrm{d}\,t$, $\mathrm{d}\,L_\mathrm{D}/\mathrm{d}\,t$ and $\mathrm{d}\,T/\mathrm{d}\,t$. When the irradiance and temperature are supplied externally, as is typically the case in coupled physical-biological models, it is not possible to obtain their temporal derivatives analytically. Hence they are numerically approximated as the

discrete backward difference between their current ($t = i$) and previous ($t = i-1$) values, divided by the integration time step, i.e., $\mathrm{d}\,E/\mathrm{d}\,t \approx (E_i - E_{i-1})/\Delta t$ for $E = \{\bar{I},\ L_\mathrm{D},\ T\}$ (see also Section 2.3.1). Finally, for the case of multiple phytoplankton functional types (PFTs) indexed by $j$, Eq. (4b.3) can be generalized as follows:

$$\frac{\mathrm{d}\,\mathrm{DIN}}{\mathrm{d}\,t} = \frac{F_{\mathrm{DON - DIN}} - \displaystyle\sum_j \mathrm{Phy_C}^j\left[\mu^j Q^j + \dfrac{\partial Q^j}{\partial \bar{I}}\dfrac{\mathrm{d}\,\bar{I}}{\mathrm{d}\,t} + \dfrac{\partial Q^j}{\partial L_\mathrm{D}}\dfrac{\mathrm{d}\,L_\mathrm{D}}{\mathrm{d}\,t} + \dfrac{\partial Q^j}{\partial T}\dfrac{\mathrm{d}\,T}{\mathrm{d}\,t}\right] - D \cdot (\mathrm{DIN} - \mathrm{DIN_{in}})}{1 + \displaystyle\sum_j \left[\mathrm{Phy_C}^j \dfrac{\partial Q^j}{\partial \mathrm{DIN}}\right]} \tag{\{IA\} (4b.4)}$$

As a technical remark regarding the FABM implementation of the model: Eqs. (4b.3)–(4b.4) require the combination of

terms which are calculated by separate abiotic and phytoplankton modules in K21. Therefore, in order to avoid a circular-dependency error in the current implementation, an intermediate module collects the necessary terms from the two modules, and sets the right-hand sides for DIN at once.



**Table 1.** Descriptions, values and units of model parameters regarding phytoplankton growth. Parameters with prime ($C'$) are for a cell with an equivalent spherical diameter (ESD) of 1μm, which is the size assumed for experiments T1 and T2. For T3, where different size classes are simulated, the respective values are obtained according to $C = C' \cdot \text{ESD}^{S_C}$, where $S_C$ is the allometric scaling coefficient for this parameter. Values for $C'$ and $S_C$ are as in Smith et al. (2016), and other parameters as in Kerimoglu et al. (2021).

| Term/Symbol | Definition | Value | Unit |
|---|---|---|---|
| $\hat{\mu}_0$ | Potential maximum growth rate | 5.0 | $\text{d}^{-1}$ |
| $Q'_0$ | Subsistence quota | 0.039 | $\text{mmolN molC}^{-1}$ |
| $S_{Q_0}$ | Allometric scaling coefficient of $Q_0$ | -0.18 | – |
| $\hat{A}'_0$ | Potential maximum nutrient affinity | 0.15 | $\text{m}^3 \text{ mmolC}^{-1} \text{ d}^{-1}$ |
| $S_{A_0}$ | Allometric scaling coefficient of $\hat{A}_0$ | -0.8 | – |
| $\hat{V}'_0$ | Potential maximum N uptake rate | 5.0 | $\text{molN molC}^{-1} \text{ d}^{-1}$ |
| $S_{V_0}$ | Allometric scaling coefficient of $\hat{V}_0$ | 0.2 | – |
| $\alpha$ | Chl-specific slope of P-I curve | 1.0 | $\text{m}^2 \text{ E}^{-1} \text{ molC gChl}^{-1}$ |
| $R_{\text{M}}^{\text{Chl}}$ | Cost of Chl maintenance | 0.1 | $\text{d}^{-1}$ |
| $\zeta_{\text{Chl}}$ | Cost of Chl synthesis | 0.5 | $\text{mmolC gChl}^{-1}$ |
| $\zeta_{\text{N}}$ | Cost of N uptake | 0.6 | $\text{molC molN}^{-1}$ |
| $m$ | Mortality rate coefficient | 0.01 | $\text{m}^3 \text{ mmolC}^{-1} \text{ d}^{-1}$ |
| $r_{\text{hyd}}$ | Hydrolysis rate constant | 0.1 | $\text{d}^{-1}$ |
| $r_{\text{rem}}$ | Remineralization rate constant | 0.1 | $\text{d}^{-1}$ |
| $D$ | Dilution rate | 0 (T1-T3), 0.1 (T4) | $\text{d}^{-1}$ |
| $\text{DIC}_{\text{in}}$ | DIC concentration in the inflow medium | 1000 | $\text{molC m}^{-3}$ |
| $\text{DIN}_{\text{in}}$ | DIN concentration in the inflow medium | 25 | $\text{molN m}^{-3}$ |
| $w_{Det}$ | Sinking rate of detritus | 0 (T1-T3); 0.2 (T4) | $\text{m d}^{-1}$ |
| $H_{\text{SML}}$ | Height of the SML | 20 | m |

## 2.3 Test setups and model operation

For all tests, the model is operated in a spatially homogeneous 1-box setup, using the 0D driver of FABM (Bruggeman and
Bolding, 2014). With this 0D setup, numerical solutions are obtained using a 4[th] order Runge-Kutta method with a time step
of 60 seconds. Model forcing applied in our 0D setup varies among different tests, as explained below.

### 2.3.1 T1

This test is designed to assess the effect of inaccuracies incurred by the numerical approximation of the time-derivatives of
external forcing variables. We consider two cases with regard to irradiance: For PAR:N the time-derivative is approximated
numerically and for PAR:A it is obtained analytically. In both cases, irradiance ($\bar{I}$) is provided (as implemented in K21) as



a sinusoidal function of day of year ($t$) to represent a seasonal cycle typical of a high latitude environment in the northern hemisphere:

$$\bar{I}(t) = \bar{I}_{\min} + (\bar{I}_{\max} - \bar{I}_{\min})0.5\left[1 + \sin\left[2\pi t'\right]\right], \qquad t' = \frac{t}{365} - 0.25 \tag{13}$$

where $\bar{I}_{\min} = 1.6\,\mathrm{mol\,m^{-2}d^{-1}}$ and $\bar{I}_{\max} = 110\,\mathrm{mol\,m^{-2}d^{-1}}$ define the minimum and maximum values throughout the year
and $t'$ represents the relative day of the year delayed by a quarter cycle to obtain the peak value at the middle of the year (see Fig. 1 for the behavior of the function with these parameters). For simplicity, we assume that temperature, T, is fixed at $10°C$, fractional day length, $L_D$, is unity, and we ignore light attenuation.

**PAR:N** in the first case, the time-derivative of $\bar{I}$ is calculated numerically as a finite-difference approximation:

$$\frac{\mathrm{d}\bar{I}}{\mathrm{d}t} \approx \frac{\bar{I}_i - \bar{I}_{i-1}}{\Delta t} \tag{14a}$$

where $i$ is the time-step index and $\Delta t$ the time step of the numerical integration.

**PAR:A** in the second case, the temporal derivative of short wave radiation is calculated analytically:

$$\frac{\mathrm{d}\bar{I}}{\mathrm{d}t} = (\bar{I}_{\max} - \bar{I}_{\min})\frac{\pi}{365}\cos\left[2\pi t'\right] \tag{14b}$$

The temporal derivatives found by the numerical and analytical approaches are almost identical (Fig. 1).

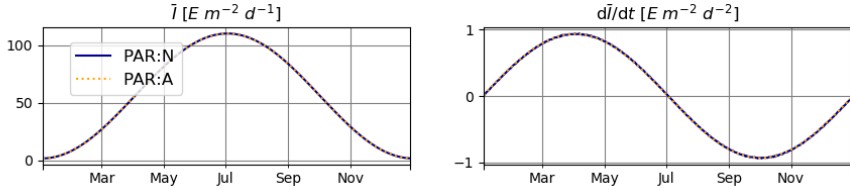

**Figure 1.** Daily average irradiance and its temporal derivative, as extracted from the simulation outputs generated for T1. PAR:N (solid blue line) is the model version where the temporal derivative of irradiance is approximated numerically; PAR:A (dashed orange line): both irradiance and its temporal derivative are calculated analytically.

### 2.3.2 T2 – T3

For these tests, we apply the numerical and analytical time-derivatives for $\bar{I}$ (Eqs. 14a and 14b) and also for variable day length, $L_D$, and temperature, T. $L_D$, as described by Forsythe et al. (1995), is used to calculate the instantaneous irradiance, based on the same irradiance function as in T1 (Eq. 13). The seasonal variability in T is represented by a sinusoidal function analogous to Eq. (13), with $T_{\min} = 2°C$ and $T_{\max} = 20°C$.

For T3, we compare simulations for 10 phytoplankton groups with the IA and DA variants. Phytoplankton groups represent
different size classes across the range of 0.2–100 µm equivalent spherical diameter (ESD), uniformly spaced on a logarithmic





scale. As in Smith et al. (2016), $Q_0$, $\hat{V}_0$ and $\hat{A}_0$ vary according to allometric relationships (Table 1). Scalings of $Q_0$ and $\hat{V}_0$ are based on a combination of cell-specific scalings of subsistence quotas (Edwards et al., 2012, 'marine species'), maximum uptake rates (Marañón et al., 2013), and cell-specific C content (Menden-Deuer and Lessard, 2000, 'protist plankton excluding diatoms'). Scaling of $\hat{A}_0$ is based on heuristics (Smith et al., 2014).

As a technical note regarding the implementation, using the `phy_Cbased.F90` and `abio_Cbased.F90` modules, the number of phytoplankton types can be modified without changing or recompiling the code, by adjusting the configuration file (see the `fabm.yaml` examples in the `testcases` folder that were employed to produce the results presented in this study), which is a feature of the modularity of FABM.

### 2.3.3 T4

In a closed system, where all mass is conserved, DIN can be calculated directly as the difference between the initial total mass and the sum of all other pools (e.g., $\mathrm{DIN} = \mathrm{Total\,N} - \mathrm{DON} - \mathrm{Det_N} - Q \cdot \mathrm{Phy_C}$). This would eliminate the neccessity of deriving the additional differentials in Eq. (4b.3) (solutions of which are provided in Appendix A), and the resulting code could be significantly faster, owing to one less state variable (i.e., DIN) and lower amounts of logic and calculations. However, this would work only for closed systems.

The aim of T4 is to evaluate the behavior of the model in open systems, such as chemostats, using a non-zero $D$ in Eqs. (1a)–(4a) to represent dilution, or the dynamics within a surface mixed layer (SML), using $D > 0$ and $w_\mathrm{Det} > 0$, to represent mixing with the layer below the SML and sedimentation of detritus out of the SML, respectively. In order to characterize an aquatic environment in a temperate climate zone that undergoes thermal stratification in summer, we consider a cyclic seasonal pattern in $D$, with values approaching to $D_\mathrm{max} = 1.0$ during winter and $D_\mathrm{min} = 0.001$ during summer:

$$D = D_\mathrm{min} + 0.25(D_\mathrm{max} - D_\mathrm{min})[1 - \sin(2\pi t')]^2, \qquad t' = \frac{t}{365} - 0.15 \tag{15}$$

where $t'$ is the relative day of the year, adjusted to mimic initiation of stratification by the beginning of April.

In order to examine the mass balance of the model in such a setup, we introduce two new state variables, $\mathrm{Ext_C}$ and $\mathrm{Ext_N}$, which trace the amounts of N and C exported from and imported into the system:

$$\frac{\mathrm{d\,Ext}_X}{\mathrm{d}t} = D \cdot (\mathrm{Phy}_X + \mathrm{Det}_X + \mathrm{DO}X + (\mathrm{DI}X - \mathrm{DI}X_\mathrm{in})) + \frac{w_\mathrm{Det}}{H_\mathrm{SML}} \cdot \mathrm{Det}_X , \quad X \in \{\mathrm{C, N}\} \tag{16}$$

such that the global amounts of N and C, i.e., the sums of these variables and the corresponding C and N variables in Eqs. (1a)–(4), should be conserved. See Table 1 for the values of the additional parameters that describe the dilution and sinking fluxes.

For this test, we consider two PFTs with ESD's of 1 and 10 $\mu$m, with $Q_0$, $\hat{V}_0$ and $\hat{A}_0$ scaled as explained for T3 above. The configuration files for this test are provided with the code (see the code availability section).



## 3 Results

<sub>185</sub> ### 3.1 Accuracy of numerical approximation of the temporal derivative of light (T1)

The model is conservative with respect to C and mostly also N (Fig. 2), for both numerically-approximated (PAR:N) and analytically-calculated (PAR:A) temporal derivatives of irradiance (see the details in Section. 2.3.1). The range of deviation (difference between maximum and minimum values obtained throughout the run) of total N is about twofold higher in the PAR:N run than in the PAR:A run (see Fig. 2), and corresponds to about 0.0003% of the total N in the system. This suggests

<sub>190</sub> that the numerical approximation of the derivative of light does introduce some additional error.

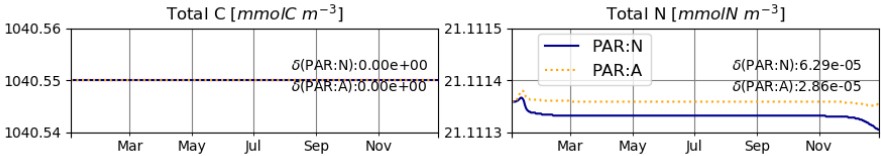

**Figure 2.** T1: Carbon (left) and nitrogen (right) pools for the PAR:N (solid blue line), and PAR:A (dashed orange line) simulations.

### 3.2 Testing the IA approach in a more realistic setup, in comparison to the fully explicit DA approach (T2)

For T2 we consider seasonal variations also in $T$ and $L_D$ (Fig. 3). Note that variations in $L_D$ also affect the seasonal cycles of $\bar{I}$ and $d\bar{I}/dt$.

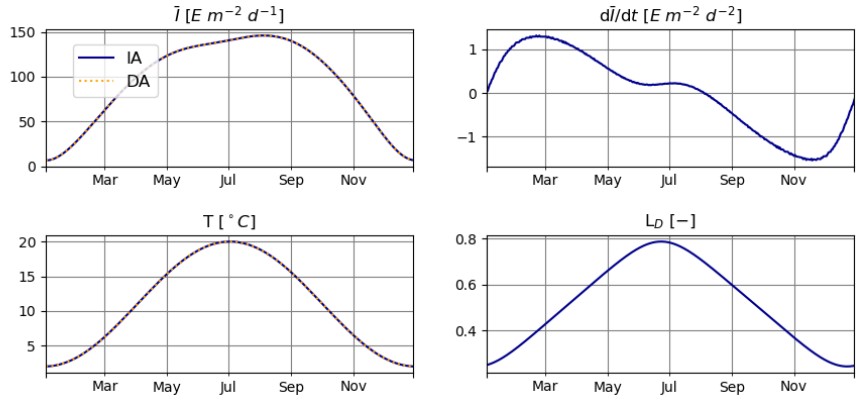

**Figure 3.** Daily average irradiance and temperature and their numerically approximated temporal derivatives used in T2.

The IA and DA variants produce almost identical results (Figs. 4, 5). This similarity is expected (Ward, 2017). On a closer

<sub>195</sub> look, some differences can be detected, such as slightly higher $Phy_N$ at the peak of the spring bloom and slightly higher $Det_N$ shortly after the spring bloom in the DA variant. The differences are due to the re-allocation of part of the fluxes between $Phy_N$ and DIN according to Eq. (4b.3). They remain relatively small because (1) the time scale of the optimal regulation of N uptake





in the DA variant is short relative to the time scale of phytoplankton growth and the DIN-changes in our simulations, and (2)
the strong interaction between phytoplankton and DIN leads to a negative feed-back between the deviations between the IA
and DA variants and the extra DIN fluxes caused by variations in $Q$ in the IA variant.

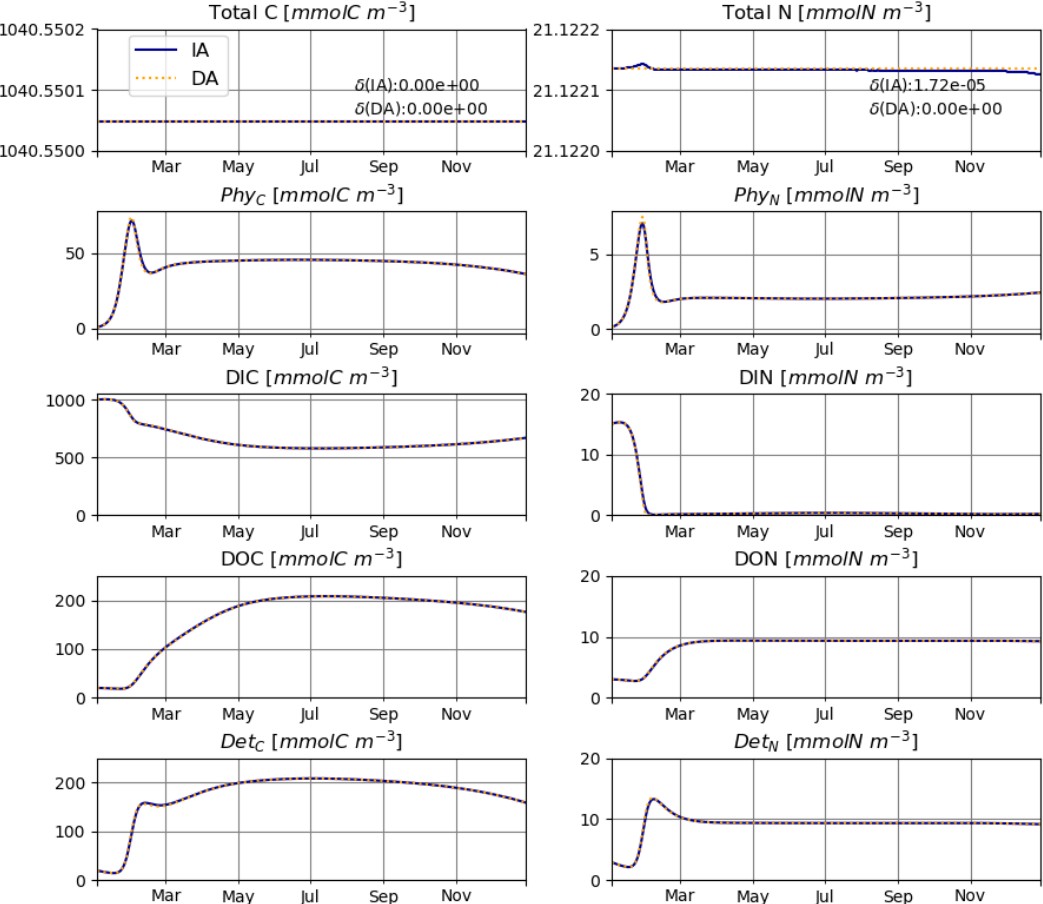

**Figure 4.** T2: Carbon (left) and nitrogen (right) pools for the IA (solid blue line) and DA (dashed orange line) variants with variable daylength
and temperature.



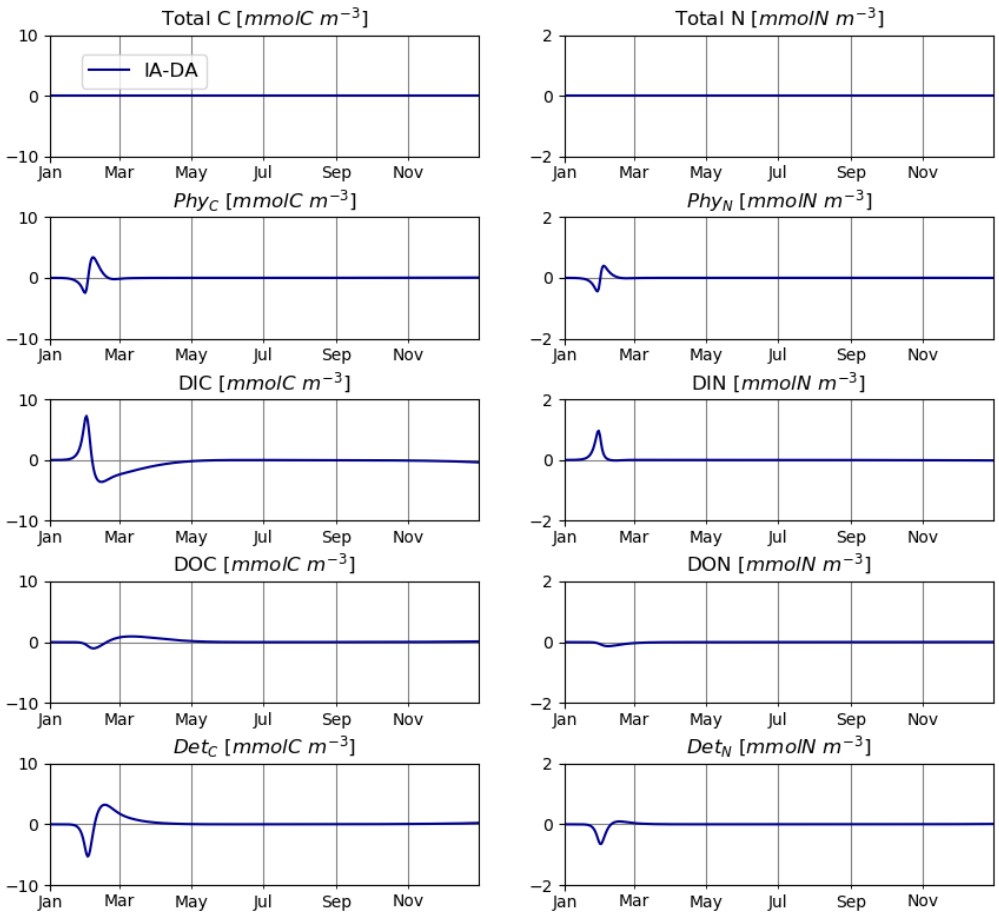

**Figure 5.** T2: Differences between the IA and DA variants for the quantities shown in Fig. 4.

## 3.3 T3: Comparing DA and IA variants in simulating in simulating multiple PFTs

Fig. 6 shows results of experiment T3 with 10 phytoplankton size classes for our IA and DA variants. Annually-averaged concentrations decrease with cell size, and the larger classes exhibit stronger seasonal relative variations. Under other environmental conditions, e.g., different initial conditions or temporal variability, different outcomes can emerge (see, e.g., Taherzadeh et al., 2017), but this is outside of the scope of our current study. C biomass of phytoplankton by the two variants is near-identical. However, differences do exist during the spring bloom, with concentrations of smaller groups being higher in the IA than in the DA variant, and vice versa for the larger groups. Concentrations of the larger groups are higher in the DA variant, likely because the extra DIN derived in the IA variant from the increasing cell quotas during the spring bloom benefits the larger cells less than the smaller cells as imposed by the allometric relationships (as in Grover, 1991; Litchman et al., 2009).





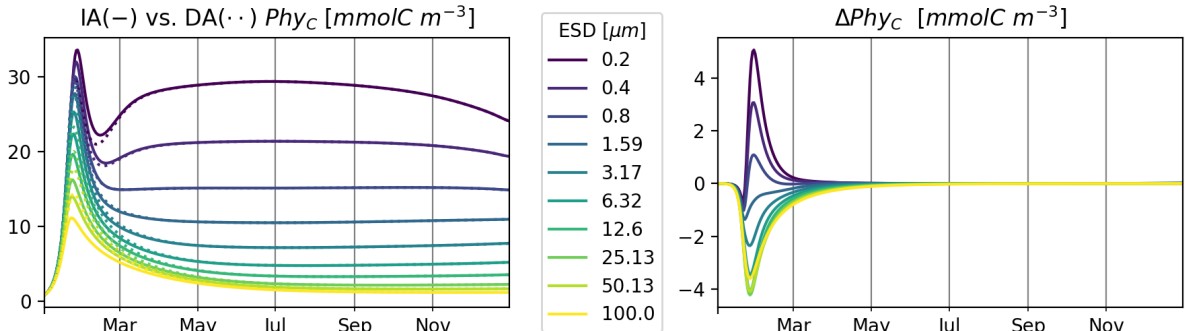

**Figure 6.** T3: C biomass of 10 phytoplankton size classes in the IA (solid line) and DA (dotted line) variants (left) and the difference between IA and DA (right).

### 3.4 T4: Comparing DA and IA variants in a non-closed system

In T4, we consider competition between two species in an open system forced by fluxes to and from an external environment. We simulate a surface-mixed-layer (SML), where mixing with the deeper layer can introduce new nutrients (DIN and DIC) and dilute all other variables, and sedimentation exports detrital C and N out of the system. As typically observed in aquatic environments located in temperate climate zones, we prescribe a seasonally-varying mixing coefficient, with lower values during summer due to thermal stratification.

Under such a regime, the system captures the characteristic features of a temperate aquatic environment, with low phytoplankton biomass during winter, a strong spring bloom, depletion of DIN within the SML during summer, and an autumn bloom. Accordingly, the total N in the system shows a strong seasonal pattern (Fig. 7). However, when the external N is taken into account, the global amount of N is consistently conserved by the model also for an open system.



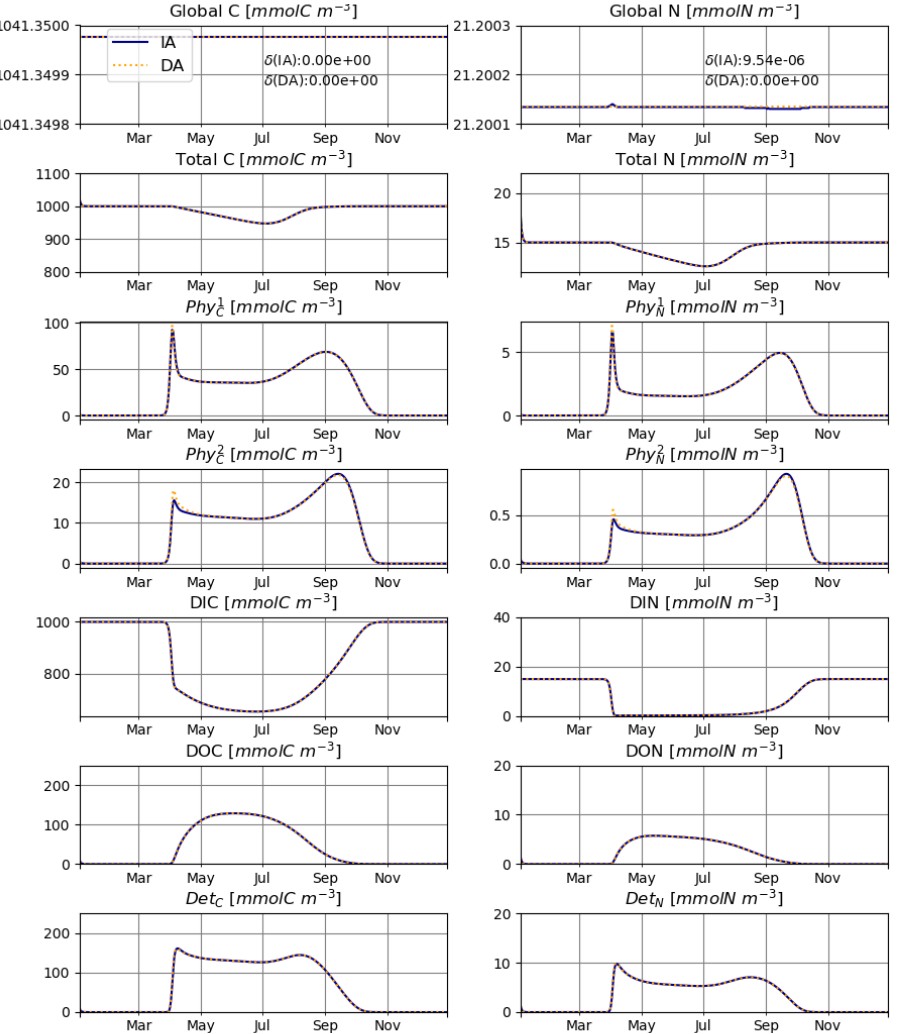

**Figure 7.** T4: Annual variations of global C and N (total C+Ext$_C$ and total N+Ext$_N$, see Eq. 16), total C and N ($\sum \mathrm{Phy}_C^j$+DIC+DOC+Det$_C$ and $\sum \mathrm{Phy}_N^j$ + DIN + DON + Det$_N$) and other state variables that trace individual C and N pools for a seasonally varying mixing regime in an open system.

## 4 Discussion

In this study, we present FABM-NflexPD 2.0, a FABM implementation of the FlexPFT model introduced by Smith et al. (2016) with a few minor corrections (see the notes at the end of Appendix A). The precursor, FABM-NflexPD 1.0 (Kerimoglu et al., 2021), resolves only the N cycle, and does not close the C-cycle. FABM-NflexPD 2.0 can resolve both N- and C-cycles in a 0D setup, owing to an additional flux term to maintain the mass balance of N (Sections 2.2 and Appendix A).



## 4.1 Re-establishing the Mass Balance

Two variants of the model are elaborated here. The 'Dynamic Acclimation' (DA) variant is fully explicit in its treatment of the C and N content of phytoplankton, as in the model by Fernández-Castro et al. (2016). The 'Instantaneous Acclimation' (IA) variant aims to track the phytoplankton dynamics with a single state variable, based on a balanced growth approximation (Burmaster, 1979), that is, assuming that cells reach the equilibrium state instantaneously.

Owing to the lack of a state variable $Phy_N$, Eqs. (1a) – (4) do not preserve mass (total N) because they ignore the contribution of the rate of change of $Q$ to the rate of change of $Phy_N$. It is impossible to maintain mass balance mechanistically without adding a state variable $Phy_N$. However, we can re-establish mass balance to a large extent, by assigning the missing N flux to the DIN compartment in Eq. (4b.3). While this may be a rather arbitrary measure for achieving N mass balance and also violates the assumptions behind the model by assigning part of $Phy_N$ to DIN, the resulting differences compared to explicitly resolving $Phy_N$ are relatively small, as was also shown previously (Ward, 2017).

Through detailed tests, we show that N is conserved to a very large degree (for all tests we conducted, max. error was 0.0063%, which was for T1). We also show that the predictions of the IA and DA variants are mostly indistinguishable. Finally, with a simulation of 10 phytoplankton size classes with our two model variants, we demonstrate that the model is well aligned with the modular coupling philosophy of FABM (Bruggeman and Bolding, 2014).

As explained in Section 2.2, reducing the errors in mass balance for N requires explicitly calculating the changes in N quota over time, i.e., $dQ/dt$, which in turn requires calculation of individual components of this change driven by different environmental factors, namely, DIN, $\bar{I}$, T and $L_D$. Under the idealized setup of T2, changes in DIN are clearly the dominant source of variation in Q. However, contributions by other factors are non-negligible (Fig. 8). In other setups, the relative importance of various environmental factors may be different. Relevance of these secondary factors to the elemental stoichiometry of phytoplankton is an often neglected aspect. We expect our mathematically explicit treatment of this issue to inspire and contribute to future endeavors to establish an analytical framework for investigating the mechanistic underpinnings of plankton physiology.

Moreover, although the model, as of its current state, is not ready to be used in a spatial setup (see below), we believe that this study can provide the basis for extension of the model for spatially explicit frameworks.



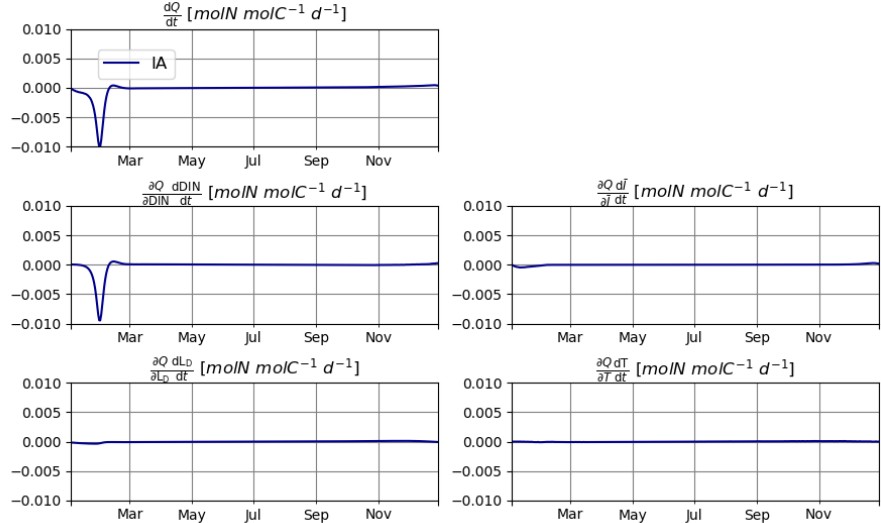

**Figure 8.** T2: Total d$Q$/d$t$ and its components as contributed by the changes in DIN, $\bar{I}$, L$_D$ and T.

## 4.2 Computational Efficiency and Application Potential

Our results demonstrate that a state variable that tracks the elemental content of plankton can be effectively removed, without leading to major issues in mass balance, in a 0D setup. In comparison to a fully explicit dynamic variant, removing a state variable does not seem to result in clear advantages in computational efficiency in such a spatially truncated setup that does not require calculation of spatial transport. Any potential reduction in computational costs owing to one less state variable in the IA variant is apparently compensated by the additional calculations required for the derivatives in Eq. (4b.3) (see Appendix-A).

The modelling framework FABM, in which the model is implemented, allows seamless coupling of models with various hydrodynamical hosts (Bruggeman and Bolding, 2014). We have also attempted an application in a 1D setup using GOTM (Burchard et al., 2006) as the hydrodynamical host, and found out that N is not conserved. This is to be expected, because the spatial transport calculations do not account for spatial gradients in $Q$, which introduces errors analogous to the difference between Eqs. (4b) and (4b.3). It may be possible to develop a mass conservative IA approach for spatially explicit models, by accounting for spatial variations of Q, in addition to its temporal variations. For FABM implementation, this would require

additional spatial flux terms to be communicated with the hydrodynamical driver. It is not clear whether the resulting model would be faster than the fully explicit variant: On one hand, in a spatially explicit setup, reducing the number of state variables would also reduce the size of the necessary transport matrices. On the other hand, fluxes associated with spatial changes in $Q$ would require additional logic and calculations. As mentioned above, we have not found evidence for performance advantages

of the IA approach in a 0D setup. However, in spatially explicit setups, transport calculations can become computationally more demanding than calculating the right-hand sides of a biogeochemical model. Therefore, reducing the number of state variables might offer computational advantages.




*Code availability.* For running the model and reproducing the results presented in this study, FABM (submodule version that matches GOTM v6.0.0) needs to be installed with its 0d driver as the 'host' (see https://github.com/fabm-model/fabm/wiki/Building-and-installing).

The version of the FABM-NflexPD used in this manuscript has been stored in a Zenodo repository, accessible under: https://doi.org/10.5281/zenodo.6600755. Instructions for compiling FABM-NflexPD for GOTM-FABM and our 0D setup are provided in README.md. The `src` folder contains the Fortran code. The model was implemented as two separate modules: the `phy.F90` module that describes phytoplankton growth and the `abio.F90` module that describes everything other than phytoplankton (See Fig. 1 in Kerimoglu et al. (2021)). The phytoplankton module can reproduce the behavior of both the IA and DA variants considered in the manuscript by setting model

parameters. The `testcases` folder contains the configuration (yaml) file that was used to produce the results for T4 in this manuscript, which can be simplified or extended to conduct the other tests.

## Appendix A: Analytical Solutions

To facilitate the solutions of the $\partial Q/\partial E$ ($E = \{\mathrm{DIN}, \bar{I}, \mathrm{T}, \mathrm{L_D}\}$) in Eq. (4b.3), we introduce a new variable $Z$ and re-write Eq. (6) in terms of $Z$ (Smith et al., 2016, S16 in the following):

$$Q = \frac{Q_0}{2}\left(1 + \sqrt{1 + \frac{1}{Z}}\right), \qquad Z = \frac{Q_0}{2}\left(\frac{\hat{\mu}_\mathrm{net}}{\hat{V}_\mathrm{N}} + \zeta_\mathrm{N}\right) \tag{A1}$$

$$\frac{\partial Q}{\partial \mathrm{DIN}} = \frac{\partial Q}{\partial Z}\frac{\partial Z}{\partial \mathrm{DIN}}, \qquad \frac{\partial Q}{\partial \bar{I}} = \frac{\partial Q}{\partial Z}\frac{\partial Z}{\partial \bar{I}}, \qquad \frac{\partial Q}{\partial L_\mathrm{D}} = \frac{\partial Q}{\partial Z}\frac{\partial Z}{\partial L_\mathrm{D}}, \qquad \frac{\partial Q}{\partial T} = \frac{\partial Q}{\partial Z}\frac{\partial Z}{\partial T} \tag{A2}$$

In Eqs. (A2), the common term $\partial Q/\partial Z$, as in S16, is:

$$\frac{\partial Q}{\partial Z} = \frac{-Q_0}{4 \cdot Z \cdot \sqrt{Z \cdot (1 + Z)}} \tag{A3}$$

Recalling $\hat{V}_\mathrm{N}$ from K21, Eq. (17):

$$\hat{V}_\mathrm{N} = \frac{(1 - f_\mathrm{A})\hat{V}_0 f_\mathrm{A}\hat{A}_0\mathrm{DIN}}{(1 - f_\mathrm{A})\hat{V}_0 + f_\mathrm{A}\hat{A}_0\mathrm{DIN}} = \frac{\hat{V}_0 \cdot \hat{A}_0 \cdot \mathrm{DIN}}{\left(\sqrt{\hat{V}_0} + \sqrt{\hat{A}_0 \cdot \mathrm{DIN}}\right)^2}, \qquad f_\mathrm{A} = \frac{1}{1 + \sqrt{\dfrac{\hat{A}_0 \cdot \mathrm{DIN}}{\hat{V}_0}}} \tag{A4}$$

We set the potential maximum rates of N and C acquisition numerically equal to the maximum-rate parameter $\mu_0$ (Pahlow et al., 2013):

$$\hat{V}_0 = \hat{\mu}_0 = \mu_0 \cdot f(T) \qquad f(T) = \exp\left[-\frac{E_\mathrm{a}}{R}\left(\frac{1}{T/\mathrm{K}} - \frac{1}{T_\mathrm{ref}/\mathrm{K}}\right)\right] \tag{A5}$$

the partial derivative of $Z$ with respect to DIN is:

$$\frac{\partial Z}{\partial \mathrm{DIN}} = \frac{\partial Z}{\partial \hat{V}_\mathrm{N}}\frac{\mathrm{d}\hat{V}_\mathrm{N}}{\mathrm{d}\mathrm{DIN}} = -\frac{\hat{\mu}_\mathrm{net} \cdot Q_0}{2 \cdot \hat{A}_0 \cdot \mathrm{DIN}^2}\left(1 + \sqrt{\frac{\hat{A}_0 \cdot \mathrm{DIN}}{\hat{V}_0}}\right) \tag{A6}$$

For calculating the partial derivative of $Z$ with respect to $\bar{I}$, $\partial Z/\partial \bar{I}$, we recall $\hat{\mu}_\mathrm{net}$, $L_\mathrm{I}$ and $\hat{\theta}$ from K21, Eqs. (20)–(23) & (26):

$$\hat{\mu}_\mathrm{net} = L_\mathrm{D}\hat{\mu}_0 L_\mathrm{I}(1 - \zeta_\mathrm{Chl}\hat{\theta}) - R^\mathrm{Chl}, \qquad R^\mathrm{Chl} = f(T) \cdot R_\mathrm{M}^\mathrm{Chl}\zeta_\mathrm{Chl}\hat{\theta} \tag{A7}$$





$$L_{\mathrm{I}} = 1 - \exp\left(\frac{-\alpha\hat{\theta}\bar{I}}{\hat{\mu}_0}\right), \qquad \hat{\theta} = \frac{1}{\zeta_{\mathrm{Chl}}} + \frac{\hat{\mu}_0}{\alpha \cdot \bar{\bar{I}}} \cdot (1 - W), \qquad W = \mathrm{W}_0\left[\left(1 + \frac{f(T) \cdot R_{\mathrm{M}}^{\mathrm{Chl}}}{L_{\mathrm{D}} \cdot \hat{\mu}_0}\right) \cdot \exp\left(1 + \frac{\alpha \cdot \bar{I}}{\hat{\mu}_0 \cdot \zeta_{\mathrm{Chl}}}\right)\right] \quad \text{(A8)}$$

where $\mathrm{W}_0$ is the 0-branch of Lambert's W-function, and $\alpha$ and $\zeta_{\mathrm{Chl}}$ are model parameters (initial Chl-specific slope of P-I curve
and cost of Chl synthesis, respectively, Table 3 in K21). $\partial Z / \partial \bar{I}$ can then be derived by canonical application of the chain rule:

$$\frac{\partial Z}{\partial \bar{I}} = \frac{\partial Z}{\partial \hat{\mu}_{\mathrm{net}}}\left(\frac{\partial \hat{\mu}_{\mathrm{net}}}{\partial \bar{I}} + \frac{\partial \hat{\mu}_{\mathrm{net}}}{\partial \hat{\theta}}\frac{\mathrm{d}\hat{\theta}}{\mathrm{d}\bar{\bar{I}}}\right) = \frac{\partial Z}{\partial \hat{\mu}_{\mathrm{net}}}\frac{\partial \hat{\mu}_{\mathrm{net}}}{\partial \bar{I}} \qquad \left(\text{because } \frac{\partial \hat{\mu}_{\mathrm{net}}}{\partial \hat{\theta}} = 0 \text{ by definition}\right) \quad \text{(A9)}$$

$$\frac{\partial Z}{\partial \hat{\mu}_{\mathrm{net}}} = \frac{Q_0}{2 \cdot \hat{V}_{\mathrm{N}}} \quad \text{(A10)}$$

$$\frac{\partial \hat{\mu}_{\mathrm{net}}}{\partial \bar{\bar{I}}} = L_{\mathrm{D}} \cdot (1 - \hat{\theta} \cdot \zeta_{\mathrm{Chl}}) \cdot \alpha \cdot \hat{\theta} \cdot (1 - L_{\mathrm{I}}) \quad \text{(A11)}$$

The day-length derivatives are

$$\frac{\partial Z}{\partial L_{\mathrm{D}}} = \frac{\partial Z}{\partial \hat{\mu}_{\mathrm{net}}}\left(\frac{\partial \hat{\mu}_{\mathrm{net}}}{\partial L_{\mathrm{D}}} + \frac{\partial \hat{\mu}_{\mathrm{net}}}{\partial \hat{\theta}}\frac{\mathrm{d}\hat{\theta}}{\mathrm{d}\bar{\bar{I}}}\right) = \frac{\partial Z}{\partial \hat{\mu}_{\mathrm{net}}}\frac{\partial \hat{\mu}_{\mathrm{net}}}{\partial L_{\mathrm{D}}} \quad \text{(A12)}$$

$$\frac{\partial \hat{\mu}_{\mathrm{net}}}{\partial L_{\mathrm{D}}} = \hat{\mu}_0 \cdot L_{\mathrm{I}} \cdot (1 - \zeta_{\mathrm{Chl}}\hat{\theta}) \quad \text{(A13)}$$

The temperature-derivative of $Z$ is obtained via the derivatives with respect to $\hat{\mu}_0$, $\hat{V}_0$ and $R^{\mathrm{Chl}}$:

$$\frac{\partial Z}{\partial T} = \frac{\partial Z}{\partial \hat{\mu}_{\mathrm{net}}}\left(\frac{\partial \hat{\mu}_{\mathrm{net}}}{\partial \hat{\mu}_0}\frac{\partial \hat{\mu}_0}{\partial T} + \frac{\partial \hat{\mu}_{\mathrm{net}}}{\partial R^{\mathrm{Chl}}}\frac{\partial R^{\mathrm{Chl}}}{\partial T}\right) + \frac{\partial Z}{\partial \hat{V}_{\mathrm{N}}}\frac{\partial \hat{V}_{\mathrm{N}}}{\partial \hat{V}_0}\frac{\partial \hat{V}_0}{\partial T}$$
$$= \left[\frac{\partial Z}{\partial \hat{\mu}_{\mathrm{net}}}\left(\frac{\partial \hat{\mu}_{\mathrm{net}}}{\partial \hat{\mu}_0} \cdot \hat{\mu}_0 - R^{\mathrm{Chl}}\right) + \frac{\partial Z}{\partial \hat{V}_0} \cdot \hat{V}_0\right]\frac{1}{f(T)}\frac{\mathrm{d}f(T)}{\mathrm{d}T} \quad \text{(A14)}$$

$$\frac{\partial \hat{\mu}_{net}}{\partial \hat{\mu}_0} = L_{\mathrm{D}} \cdot (1 - \zeta_{\mathrm{Chl}}\hat{\theta})\left[L_{\mathrm{I}} - (1 - L_{\mathrm{I}})\frac{\alpha \cdot \bar{I}}{\hat{\mu}_0}\hat{\theta}\right] \qquad \frac{\partial Z}{\partial \hat{V}_0} = -\hat{\mu}_{\mathrm{net}}\frac{Q_0}{2\hat{\mu}_0\sqrt{\hat{\mu}_0 \cdot \hat{V}_{\mathrm{N}}}} \quad \text{(A15)}$$

$$\frac{1}{f(T)}\frac{\mathrm{d}f(T)}{\mathrm{d}T} = \frac{E_{\mathrm{a}}}{R \cdot (T/\mathrm{K})^2} \quad \text{(A16)}$$

We would like to clarify that (1) replacement of $\hat{\mu}_g$ (in S16, $\hat{\mu}^I$) with $\hat{\mu}_{\mathrm{net}}$ in our model (see K21) results in the appearance
of $(1 - \hat{\theta} \cdot \zeta_{\mathrm{Chl}})$ when computing $\partial \hat{\mu}_{\mathrm{net}}/\partial \bar{I}$; (2) $L_{\mathrm{D}}$ used to be implicit in S16, now it's explicit (K21 Eq. 21) therefore it appears
for $\partial \hat{\mu}_{\mathrm{net}}/\partial \bar{I}$ unlike in S16; (3) Changes in $Q$ due to T were not accounted for by Smith et al. (2016).

*Author contributions.* OK and SLS conceived and designed the study with contributions from MP; OK and MP extended the model with
fluxes required to satisfy the mass balance; OK implemented the model in FABM; PA configured and performed the runs with multiple PFTs;
OK drafted the manuscript; MP and SLS contributed to writing and revision of the text.

*Acknowledgements.* This research has been supported by the German Research Foundation, DFG (grant no. KE 1970/2-1, PI: OK) and the
Japan Society for the Promotion of Science, JSPS (PI: SLS). We acknowledge the developers of the open-source software used in this study,
foremost FABM and GOTM.



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
