# Peer review of "FABM-NflexPD 2.0: Testing an Instantaneous Acclimation Approach for Modelling the Implications of Phytoplankton Eco-physiology for the Carbon and Nutrient cycles"

_EGUsphere, 2022_

## Author Response (AR1)

**Response to Reviewers**

**Reviewer #1**

We thank the reviewer for the positive evaluation. Below, the original comments of the reviewer are quoted verbatim, and our responses are provided right after each comment. Modified or newly added text are provided in blue fonts.

*"The paper presented a model of phytoplankton instantaneous acclimation (IA), FABM-NflexPD 2.0 (K21), which was further developed from its earlier version FABM-NflexPD 1.0 (K11). The K21 was extended from K11 to account for and conserve both C and N fluxes. The K21 was said might lower in computational costs compared to its DA (Dynamic Acclimation) variant due to less state variables. The K21 was then tested in 4 scenarios and its performance was compared to its DA variant.*
Response: To prevent confusion, we would like to clarify that the abbreviation K21 stands for the earlier version of the model that was described by Kerimoglu et al. 2021, whereas we refer to the new version described by this current manuscript as K22. To make this point clearer, we reformulated the opening paragraph of Discussion (L221-225) as follows:
"In this study, we present FABM-NflexPD 2.0, a FABM implementation of the FlexPFT model introduced by Smith et al. (2016) with a few minor corrections (see the notes at the end of Appendix A). The precursor, FABM-NflexPD 1.0 (K21 Kerimoglu et al., 2021), resolves only the N cycle, and does not close the C-cycle. FABM-NflexPD 2.0, which we present here, can resolve both N- and C-cycles in a 0D setup, owing to an additional flux term to maintain the mass balance of N (Sections 2.2 and Appendix A)."

*The paper has achieved its goals, e.g., the model was successfully built and its performance was almost as same as the DA's. However, the treatment of N mass balance (section 4.1) that made it violate the model assumption sounded unconvinced. Has the paper tried out alternative treatments to this issue?*
Response: The assumption violated by our IA formulation is that the N flux from DIN to phytoplankton ends up as $Phy_N$, as we have to assign part of $Phy_N$ to DIN in order to maintain mass balance for total N.  While it would be possible to assign (part of) the missing fluxes to other tracers (e.g., detritus N), this would still violate the assumptions underlying the model formulation.  Unfortunately, therefore, it is indeed impossible to maintain mass balance without either violating model assumptions or having the $Phy_N$ tracer in the model. However, it is one of the goals of our ms to point out explicitly this kind of problem involved in attempting to simplify biogeochemical models.

*Might the paper state strength, weakness and applications of the K21?"*
Response: we included a new conclusion section to summarize the take home messages of the study, that includes what was achieved and what is left for future work, as follows:
Accounting for the variability in phytoplankton cellular composition is required for a realistic representation of nutrient cycling. Variable cellular composition is usually described by a Dynamic Acclimation (DA) approach, which requires additional state variables for the cellular constituents, thereby increasing computational costs. Smith et al. (2016) proposed the Instantaneous Acclimation (IA) approach, which approximates the variability in cellular composition without the need for additional state variables. As long as only one of carbon (C) or nitrogen (N) is resolved (i.e., the mass balance is closed globally), the IA approach is fully conservative and can be applied, e.g., for ecologically oriented questions (e.g. Kerimoglu et al.,

2021). Here we provide a formally consistent and complete explanation of how the mass balance can be nearly re-established for when both N and C are resolved by an IA model. Through several tests in 0D setups, we demonstrate that under stable environmental conditions, the fully explicit model can be closely reproduced, but that transient differences between the IA and DA variants can emerge and mass balance can be slightly compromised. A generalization of the IA approach to account also for spatial variability will require extending our (0D) IA framework towards spatially explicit setups. In our 0D setup, we did not find evidence for improved computational efficiency. However, gains in spatially explicit setups may be possible, given that the number of state variables to be transported is known to significantly affect the computational costs.

*"Technical errors: (1) line 14: approach; (2) p.11, title of 3.3: "in simulating" appeared twice."*
Response: We would like to thank the reviewer for pointing to these technical errors, which are now corrected.

**Reviewer #2**

We thank the reviewer for the review and suggestions. Below, the original comments of the reviewer are quoted verbatim, and our responses are provided right after each comment.

*"Kerimoglu and colleagues developed an updated version of their previously published plankton model FABM-NflexPD. In this version, they track both N and C biomass of phytoplankton assuming instantaneous acclimation (IA version of the model). In comparison to its previous version, this new version conserves both carbon and nitrogen in the system. Mass balance is ensured by analytically computing the temporal change in cellular N quota. In 0-D and pseudo 0-D setups, mass conservation is excellent and the model performs very well compared to a fully explicit treatment of the N quota (DA version of the model). However, the IA setup is not cheaper in terms of computing cost. The paper is very well written, very clear and complete. I don't have any major issues on what is presented in the study.*
Response: we thank the reviewer for this positive assessment.

*"However, I should admit that I have trouble finding this paper interesting and useful. The main objective of this study, as stated by the authors, is to develop a model that mimics the behavior of a full quota model but that is cheaper so that it can be embedded in a global biogeochemical model. As a global biogeochemical modeller, I agree that it is a crucial point. And having less tracers in a global 3-D model is generally a good strategy to reduce the computing cost as transport of a tracer is very expensive. In the case of this study, I think that this main objective is not reached."*
Response: we agree with the reviewer that we did not reach that ultimate goal (of building a more efficient model) that motivated this study. In fact, the specific objective of this study, as stated in the final paragraph of the introduction (L47-48) was to ' evaluate the consistency and robustness of the [newly developed] model by means of ... formal tests'. We believe that our work has reached this specific objective and will also be potentially useful in reaching the aforementioned ultimate goal, by the virtues of 1) laying out the mass balance problem on a more formal basis; 2) explaining how this mass balance problem can be repaired, but that this repair is partial and not exact; 3) how the various dependencies of quota on external factors should be taken into account; 4) setting a framework to test this approach, and pointing to various aspects that need to be taken into account in such a complex photoacclimation approach with both direct and indirect dependencies. We would like to point out the fact that all these outcomes are novel. We introduced a conclusion section to summarize the specific objectives and take home messages of this study, and the remaining challenges to achieve an efficient model that can mimic the expensive full quota model as follows:

Accounting for the variability in phytoplankton cellular composition is required for a realistic representation of nutrient cycling. Variable cellular composition is usually described by a Dynamic Acclimation (DA) approach, which requires additional state variables for the cellular constituents, thereby increasing computational costs. Smith et al. (2016) proposed the Instantaneous Acclimation (IA) approach, which approximates the variability in cellular composition without the need for additional state variables. As long as only one of carbon (C) or nitrogen (N) is resolved (i.e., the mass balance is closed globally), the IA approach is fully conservative and can be applied, e.g., for ecologically oriented questions (e.g. Kerimoglu et al., 2021). Here we provide a formally consistent and complete explanation of how the mass balance can be nearly re-established for when both N and C are resolved by an IA model. Through several tests in 0D setups, we demonstrate that under stable environmental conditions, the fully explicit model can be closely reproduced, but that transient differences between the IA and DA variants can emerge and mass balance can be slightly compromised. A generalization of the IA approach to account also for spatial variability will require extending our (0D) IA

framework towards spatially explicit setups. In our 0D setup, we did not find evidence for improved computational efficiency. However, gains in spatially explicit setups may be possible, given that the number of state variables to be transported is known to significantly affect the computational costs.

*"First, the study is restricted to a pseudo 0-D (closed and opened) framework where transport with neighboring cells is not relevant and computing not an issue. Second, they claim that transposing this framework to a 1-D setup failed because mass is no more conserved. Obviously, spatial transport of a variable quota leads to the same problem as temporal evolution of this quota. As said in the manuscript, conserving mass in a 1-D or 3-D configuration would require to track the evolution of the quota due to transport to compute the additional fluxes of nutrients. To me, this is equivalent to explicitly transport the quota."*
Response: More precisely, we had stated that 'It may be possible to develop a mass conservative IA approach for spatially explicit models, by accounting for spatial variations of Q, in addition to its temporal variations' (L259-260). Spatial transport of a variable quota indeed introduces problems which are qualitatively different from those due to the temporal variations considered here, similar to the differences between solving ordinary and partial differential equations. Transport and mixing schemes, e.g., for convective overturning or along-isopycnal mixing, often involve transport across several grid cells in one time step. Also, in FABM and all 3D ocean models known to us, the biogeochemical modules cannot see tracers in other grid cells. Thus, accounting for spatial quota variations must follow a different approach, but until such an approach will be developed, the question should be considered open whether it will be equivalent (also in terms of computational cost) to explicitly transporting the quota.

*"Furthermore, it would require additional fluxes of nutrient that could possibly, especially when transport and spatial gradients are strong, significantly alter the model behavior. In other words, the computing cost would be identical for a result that may differ from the fully explicit model."*
Response: This is indeed a potential outcome, but is subject to future research.

*"I have additional small questions. In T1, mass is not fully conserved in both model versions. Could the authors be more specific on why this is the case? In DA, is it simply truncation errors in single precision?"*
Response: We believe that the N mass imbalance shown in Fig. 2 is indeed due to the limited precision of the numerical solution because it depends strongly on the method of numerical integration. Also, we would like to clarify that in T1, the comparison is not between the IA and DA variants, but between two IA variants, where the temporal derivative of irradiance is approximated numerically (PAR:N) vs. calculated analytically (PAR: A), for the sake of quantifying the additional error introduced by the numerical approximation of the unknown derivative of irradiance in a more realistic setup, where irradiance is not described analytically, but as external forcing.

*"In T2, differences in total N seem to be 0? Obviously, this is not exactly 0 because it is not the case in T1. Is the difference larger than in T1? In other words, I suggest to change the y-axis in a way similar to what is done in T1.*
Response: Thanks for this suggestion. In Fig. 5, we have changed the y-axis for Total N to make the differences more visible.

*Finally, in T3, the authors only show two figures. From these figures, it is difficult to see if the temporal evolution of the total phytoplankton biomass is changed and by how much.*
Response: Thanks again for this suggestion. We have amended this figure (Fig. 6) with the total phytoplankton biomass.

*To conclude, I find that the authors did not make the demonstration that the framework they developed in the study provides an interesting, cheaper alternative to model flexible nutrient quota in spatially explicit biogeochemical models. To be convinced, I think that including a 1-D experiment is necessary. This would also present how additional fluxes due to transport can be represented and if that framework is really cheaper than a full model. Without such an experiment, I think that this manuscript should be rejected."*

Response: As is clearly explained in section 4.2, in its current state, the model is indeed not ready to be used in spatially explicit biogeochemical setups. However, we maintain the view that extension of the model to a spatially explicit framework should be addressed in a separate study, as this poses formidable challenges and will require its own approach and tests. As stated above, we believe that the current study constitutes substantial progress and will be useful in reaching the end goal of developing an efficient model that mimics the full quota model. We also believe that the novelties presented in this work, as listed above, deserve publication. Finally, by exemplifying a transparent and objective evaluation of a newly developed biogeochemical model component, fitting well within the scope of GMD, we believe that publication of this work will encourage other researchers to also thoroughly evaluate their models and communicate their weaknesses transparently.